# Emissions of NO, NO$_2$, and PM from inland shipping

Ralf Kurtenbach, Kai Vaupel, Jörg Kleffmann, Ulrich Klenk, Eberhard Schmidt and Peter Wiesen

Bergische Universität Wuppertal, Institute for Atmospheric and Environmental Research, 42097 Wuppertal / Germany

*Correspondence to*: Ralf Kurtenbach (kurtenba@uni-wuppertal.de)

**Abstract.** Particulate matter (PM) and nitrogen oxides NO$_x$ (NO$_x$ = NO$_2$ + NO) are key species for urban air quality in Europe and are emitted by mobile sources. According to European recommendations, a significant fraction of road freight should be shifted to waterborne transport in the future. In order to better consider this changed emission pattern in future
emission inventories, in the present study, inland water transport emissions of NO$_x$, CO$_2$ and PM were investigated under real world conditions at the river Rhine, Germany in 2013. An average NO$_2$/NO$_x$ emission ratio of 0.08 ± 0.02 was obtained, which is indicative of ship diesel engines without exhaust gas after-treatment systems. For all measured motor ship types and operation conditions overall weighted average emission indices as emitted mass of pollutant per kg burnt fuel of EI$_{NO_x}$ = 54 ± 4 g kg$^{-1}$ and a lower limit EI$_{PM_1}$ = ≥ 2.0 ± 0.3 g kg$^{-1}$ were obtained. EIs for NO$_x$ and PM$_1$ were found to be in the range of
20–161 g kg$^{-1}$ and ≥ 0.2–8.1 g kg$^{-1}$, respectively. A comparison with threshold values of national German guidelines shows that the NO$_x$ emissions of all investigated motor ship types are above the threshold values, while the obtained lower limit PM$_1$ emissions just within. To reduce NO$_x$ emissions to acceptable values, implementation of exhaust gas after-treatment systems is recommended.

## 1 Introduction

Particulate matter (PM) and nitrogen dioxide (NO$_2$) are key species for urban air quality in Europe. Whereas the exceedence of PM limiting values has attracted considerable public attention during the last decade, NO$_2$ is a topical problem, which became mature through the introduction of new European limiting values in January 2010.

The reduction of nitrogen oxide (NO$_x$ = nitrogen monoxide (NO) + NO$_2$) emissions has been historically one of the key objectives for improving air quality in Europe. NO$_x$ emissions have started to decrease considerably since the mid eighties of
the last century in many European areas. However, emissions from mobile sources are still important contributors to air pollution, in particular for NO$_x$. Together with NO$_x$, non-methane volatile organic compounds (NMVOCs) undergo photochemical reactions producing secondary pollutants such as ozone (O$_3$), peroxyacetyl nitrate (PAN) and others (Chameides et al., 1997, Atkinson, 2000).

According to the European Commission's White Paper (2011), 30 % of road freight transported over more than 300 km distance should shift to other transport modes such as waterborne or rail transport by 2030, and more than 50 % by 2050 (European Commission, 2011). Accordingly, such a shift will result in an increase of emissions from inland water transportation in the next years.

Today in Germany the contribution of inland navigation to the total freight traffic is about 12 % (BDA, 2015a). In the Rhine corridor the contribution is 16-18 %, respectively (BDA, 2015b). With respect to the goods categories "coal, crude oil and petroleum gas", "ores, industrial rocks and minerals, other mining products" and "coking plant and petroleum products" inland water navigation is the most important transportation mode. In comparison to road transport, inland navigation has a contribution of 72 % for these goods categories and 52 % for container transport. Inland water navigation is a competitive

alternative to road and rail transport, because the energy consumption per km and ton of transported goods is only approximately 17 % of road and 50 % of rail transport (ECT, 2015). As a consequence of the lower energy consumption, inland water transportation emits significantly less $CO_2$ and, therefore, has a direct impact on climate change.

In the European Union the emission of $NO_x$, VOC, PM and CO from road and rail transport decreased from 1990 to 2000, whereas emissions from inland navigation remained more or less constant and emissions from sea transport slightly

increased (Trends, 2003). However, in the Netherlands a slight reduction in inland shipping emissions were observed in the same time period when modern engines were introduced in the fleet (CTRC, 2003).

It has been also conclusively demonstrated that the fuel has an important impact on the emissions. Using liquid natural gas (LNG) as fuel for inland water vessels leads to substantial emission reductions, i.e. 75 % for $NO_x$, 97 % for PM and 10 % for $CO_2$ (Van der Werf, 2013).

The emissions from inland water transportation have been regulated by several national and international guidelines. In 2005 the German national guideline "Binnenschiffabgasverordnung, BinSchAbgasV" was implemented for national water ways, defining engine dependent emission indices, i.e. emitted mass of pollutant per kg burnt fuel, for $NO_x$ and PM of $EI_{NO_x}$: 30-42 g $kg^{-1}$ and $EI_{PM}$: 1.2-2.4 g $kg^{-1}$, respectively (BinSchAbgasV, 2005). In 2011 an international guideline for the Rhine river "RhineSchUO" was implemented with engine dependent $EI_{NO_x}$: 28-36 g $kg^{-1}$ and an $EI_{PM}$: 0.9-3.1 g $kg^{-1}$ (RheinSchUO,

2011). In addition, for river-sea-ships the MARPOL guideline (International Convention for the Prevention of Pollution from Ships) (IMO, 2012) has to be applied. For example, for marine diesel engine with a medium-speed of 720 $min^{-1}$ $NO_x$-emission indices of 58 g $kg^{-1}$ since 2000 (Tier I), 56 g $kg^{-1}$ since 2011 (Tier II) and 11 g $kg^{-1}$ since 2016 (Tier III) have been introduced.

The correct determination of emission indices (EI) is prerequisite for establishing and developing emission inventories

(VBD, 2001, Klimont et al., 2002, Browning and Bailey, 2006, Rohacs and Simongati, 2007, TNO, 2008, CBS, 2009, UBA, 2013). Up to now, several studies have been published in which NO, $NO_2$, $SO_2$ and PM emissions from sea ships (Sinha et al., 2003, Chen et al., 2005, Eyring et al., 2005, Petzold et al., 2008, Moldanova et al., 2009, Murphy et al., 2009, Schrooten et al., 2009, Williams et al., 2009, Eyring et al., 2010, Beecken et al., 2014, Jonsson et al., 2011, Lack et al., 2011, Alfödy et al., 2013) and, in particular, from sea ferries (Cooper et al., 1996, 1999, Copper, 2001, 2003, Copper and Ekström, 2005,

Tzannatos, 2010, Pirjola et al., 2014) were investigated. Motor test bed studies can also be used for the determination of EIs from single ship's engines (Petzold et al., 2008). However, up to now only three studies have reported on inland water transportation emissions (Trozzi and Vaccaro, 1998, Kesgin and Vardar, 2001, Schweighofer and Blaauw, 2009, Van der Gon and Hulskotte, 2010)

In the present study, inland water transport emissions were investigated under real world conditions at the riverside of the river Rhine in Germany during a field campaign from February 20, to February 22, 2013.

## 2 Description of the Experimental Procedures

### 2.1 Measurement site

The measurement campaign was carried out at the river Rhine in Germany close to the "Wunderland Kalkar" at Rhine
kilometre 843. Figure 1 shows a map of the measurement site. During the campaign emissions from both, upstream and downstream cruising inland ships were studied. The sampling point was located 50 m downwind from the river bank.

It is reasonable to assume that the engines of the ships passing the sampling site, were under warm operation conditions.

### 2.1 Analytical Equipment

The analytical equipment used was installed in a mobile van with an external power supply. NO and $NO_2$ were measured on-
line with a commercial $NO_x$ chemiluminescence analyzer (Environnemental, AC 31M with molybdenum converter). The time resolution was 10 s and the detection limit, which was calculated from the variation of the zero signal was 2 ppbV for NO and 3 ppbv for $NO_2$. The NO channel of instrument was directly calibrated by diluted standard NO calibration mixtures (Messer, stated accuracy 5 %). The $NO_2$ channel was calibrated by using a NO titration unit (Environnemental, GPT). $NO_2$ was produced by the reaction of NO with $O_3$ in a flow reactor leading to the quantitative conversion of the calibrated NO
($\Delta NO = \Delta NO_2$).

Ozone ($O_3$) was measured on-line with a commercial $O_3$ monitor (Environnemental, O3 41M with UV absorption). The time resolution was 10 s and the detection limit, which was calculated from the variation of zero measurements, was 1 ppbv. $O_3$ was calibrated by using an $O_3$ calibration unit (Environnemental, K-$O_3$, accuracy 10 %). $O_3$ was produced by the photolysis of synthetic air in a flow reactor leading to the quantitative formation of $O_3$.

Carbon dioxide ($CO_2$) was measured on-line with a commercial $CO_2$ monitor (LICOR 7100 with IR absorption). The time resolution was 1 s and the detection limit, which was calculated from the variation of zero measurements, was 0.5 ppmv. $CO_2$ was directly calibrated by diluted standard $CO_2$ calibration mixtures (Messer, stated accuracy 2 %).

PM was measured by an optical particle counter (OPC) (Grimm Aerosol Technik GmbH, DustMonitor EDM 107). The OPC counts particles in a size range from 0.25-32 µm in 31 size-channels. The time resolution was 6 s and the detection limit 0.1
µg m$^{-3}$. However, the instrument only provided the concentrations of the fractions $PM_1$, $PM_{2.5}$ and $PM_{10}$.

Meteorological parameters, such as temperature, pressure, relative humidity and wind speed were also measured. In addition to the measurement of compounds in the ambient air, the number and types of ships passing the measurement site were counted.

Samples were taken at a height of about 3 m above the stream gauge of the river Rhine.

## 3. Results and Discussion

### 3.1 Inland water transportation emissions

NO, $NO_2$, $O_3$, $CO_2$, $PM_1$ and $PM_{10}$ concentrations, wind speed and wind direction at the measurement site as well as movements of the ships were measured. During the campaign more than 170 emission peaks from motor ships were observed. From these peaks almost 140 could be attributed to single ships types (G=goods ship, T=petroleum tanker, PT=push tow) and were analyzed accordingly. Figure 2 shows as an example the temporal variation of NO, $NO_2$, $O_3$ and $CO_2$ mixing ratios at the measurement site on February 20, 2013 from 11:30 to 14:00. The perfect correlation between NO and $NO_2$ with $CO_2$ confirms that these compounds were emitted from the same source, i.e. the engine exhaust. The anti-correlation between $NO_2$ and $O_3$ provides information about $NO_x$ chemistry in the ship exhaust plumes, i.e. the formation of $NO_2$ by the titration reaction of NO with $O_3$.

### 3.2 $NO_2/NO_x$ emission ratio

In order to obtain information about the ships engine types and to estimate the impact of ship emissions on the ozone formation the $NO_2/NO_x$ ratio in the exhaust plume is an important parameter. It is well known that diesel engines without after-treatment systems show $NO_2/NO_x$ ratios of 0.10-0.12 for road traffic (Kurtenbach et al., 2001, Kousoulidou et al., 2008, Carslaw and Rhys-Tyler, 2013) and (0.14±0.04) for navigation (Cooper, 2001, Grice et al., 2009). In contrast, the $NO_2/NO_x$ ratio from road traffic diesel engines with after-treatment systems such as oxidation catalyst or PM filter systems are in the range of 0.25-0.30. The $NO_2/NO_x$ emission ratio from navigation diesel engines with selective catalytic $NO_x$ reduction systems (SCR) is (0.009±0.003) (Cooper, 2001).

To obtain the correct $NO_2/NO_x$ emission ratio from the measurements it is important to distinguish between primarily emitted $NO_2$ and $NO_2$, which is being formed by the reaction of NO with ozone in the exhaust plume. The correct $NO_2/NO_x$ ratio is obtained by plotting $O_x$, which is the sum of $NO_2$ and $O_3$ versus the measured $NO_x$ concentration as shown in Figure 3 (Clapp and Jenkin, 2001). The $NO_2/NO_x$ emission ratio and the local $O_3$ background mixing ratio are obtained from the slope and intercept of the regression line, respectively. From the data shown in Fig. 3 a $NO_2/NO_x$ emission ratio of (0.08±0.02) and a local ozone background volume mixing ratio of (23±2) ppbv were obtained. The obtained $NO_2/NO_x$ ratio indicate that the ships passing the measurement site were equipped with conventional diesel engines without exhaust after-treatment.

## 3.3 $PM_1$ and $PM_{10}$ emissions

Figure 4 shows the temporal variation of $CO_2$, $PM_{10}$ and $PM_1$ concentrations at the measurement site on February 20, 2013 from 11:50 to 12:10. Some $PM_1$ peaks are well correlated with those of $CO_2$ mixing ratios, therefore, with ship plumes. In contrast, some $PM_{10}$ peaks showed no correlation with ship emissions. This indicates that the main PM emissions from ships
diesel engines are in the $PM_1$ range. This result is in good agreement with other studies e.g. from the US-EPA (1996), Petzold et al. (2008), Beecken et al. (2014), Pirjola et al. (2014) and Westerlund et al. (2015). Therefore, in the present study particle ship emissions are defined as $PM_1$. According to Westerlund et al. (2015) the maximum in the particle number size distribution was observed at about 10 nm and the maximum particle mass distribution at 250 nm. Therefore the used optical particle counter (OPC) detect only a lower limit of the emitted particle mass.

**3.4 Emission indices**

From the measurement data, emission indices (EIs) for $NO_x$ (NO calculated as $NO_2$) and $PM_1$ (unit: mass per kg burnt fuel) were calculated. In Figure 5 the integrated emission peak (peak area) for NO, $NO_2$, $CO_2$ and $PM_1$ as $\Delta NO$, $\Delta NO_2$, $\Delta CO_2$ and $\Delta PM_1$ are shown as an example for a single motor ship. If one assumes that the increase of NO, $NO_2$, $PM_1$ and $CO_2$ in the plume is proportional to the emission strength of the ship engine, an emission ratio to $CO_2$, e.g. $\Delta NO_x/\Delta CO_2$, can be easily
calculated (Petzold et al., 2008). In addition the $\Delta NO$, $\Delta NO_2$, $\Delta CO_2$ and $\Delta PM_1$ were also calculated by the difference between background and plume mixing ratios (Schlager et al., 2008) and considering the precision errors of the background data of typically ±2 ppbv, ±4 ppbv, ±2 ppbv, ±1 ppmv and ±2 µg m$^{-3}$ for NO, $NO_2$, $O_3$, $CO_2$ and $PM_1$, respectively.

Both approaches were used to calculate the emission indices and were in good agreement, in general better than ±6 %. Caused by the slightly different time responses of the instruments, finally the integrated peaks results were specified.
Elementary analysis of a typical ship diesel fuel yielded: 86 wt% carbon and 14 wt% hydrogen (Cooper, 2001). From the wt% carbon and under the assumption that all fuel is burnt to the final end product $CO_2$ an emission index EI ($CO_2$) of 3,150 g $CO_2$ per kg burnt fuel was calculated and further used to calculate the corresponding emission index (EI) for the ship engines. The emission index (EI) is calculated by the following equation (1) (Petzold et al., 2008):

$$ EI(X) = EI(CO_2) \times \frac{M(X)}{M(CO_2)} \times \frac{\Delta(X)}{\Delta(CO_2)} , \qquad\qquad (1) $$

where $M$ denotes the molecular weight and $\Delta$ the peak area, mixing ratios, column densities, etc. of the species. M ($CO_2$) with 44 g mol$^{-1}$ and M ($NO_x$) with 46 g mol$^{-1}$, $NO_x$ as $NO_2$ were used for the subsequent calculations. Table 1S of the supplement summarizes the calculated EIs of the different ship types and operation conditions. Errors were calculated using error propagation for the different measured compounds.

Figure 6 shows as an example the emission index for $NO_x$ (as $NO_2$) ($EI_{NO_x}$) of single motor ships [goods] and the weighted average $EI_{NO_x}$ for different operation parameters, i.e. L=loaded, U=unloaded, A=upstream and D=downstream.

Figure 7 shows as an example the obtained lower limit $PM_1$ emission index ($EI_{PM_1}$) for single motor ships [goods] and the weighted average $EI_{PM_1}$ for different operation parameters, i.e. L=loaded, U=unloaded, A=upstream and D=downstream.

Red bars show outliers ($4\sigma$ limit) and were not taken into account in he calculation of the weighted average value. Values are lower limits because of the detection range of the OPC system.

Although Fig. 6 and 7 show a large variation of the EIs for $NO_x$ and $PM_1$, the average data exhibit that the $EI_{NO_x}$ are almost independent of engine operation parameters within the given error limits. The same was found for tankers and push tows, see weighted average emission index figures 8 and 9.

Figure 8 exhibits that the $NO_x$ emission indices of all motor ship types investigated are above the engine rotation speed dependent limit values of the German guide lines, which are 29-37 g $kg^{-1}$ for the RheinSchUO and 36-46 g $kg^{-1}$ for the BinSchAbgasV guidelines.

Figure 9 exhibits that the obtained lower limit $PM_1$ emissions values for almost all motor ship types are just within the limit values of the German guide lines, which are 0.9-3.1 g $kg^{-1}$ for the RheinSchUO and 1.2-2.4 g $kg^{-1}$ for the BinSchAbgasV

guide lines depending on the engine rotation speed.

For comparison with literature data, uncertainty($2\sigma$)-weighted averaged $EI_{NO_x}$ and $EI_{PM_1}$ were calculated for all motor ship types and operation condition investigated. An $EI_{NO_x}$ of $52\pm3$ g $kg^{-1}$ and a lower limit $EI_{PM_1}$ of $\geq 1.9\pm0.3$ g $kg^{-1}$ were obtained. Minimum and maximum EIs for $NO_x$ and $PM_1$ were found to be in the range of 20-161 g $kg^{-1}$ and $\geq$ 0.2-8.1 g $kg^{-1}$, respectively. Table 1 show the emission indices $NO_x$ and $PM_1$ in g $kg^{-1}$ fuel calculated from the measured values in

comparison with different literature data. Errors were calculated using error propagation for the different measured compounds.

Between 1998 and 2013 only a few studies reported $EI_{NO_x}$ and $EI_{PM_1}$ from inland water navigation (Trozzi and Vaccaro, 1998, Kesgin and Vardar, 2001, Schweighofer and Blaauw, 2009, Van der Gon and Hulskotte, 2010) in the range 39-57 g $kg^{-1}$ and 0.7-1.9 g $kg^{-1}$, respectively, see table 1. The uncertainty($2\sigma$)-weighted averaged $EI_{NO_x}$ and $EI_{PM_1}$ were $48\pm4$ g $kg^{-1}$

and $EI_{PM1}$ $1.3\pm0.2$ g $kg^{-1}$, which are in good agreement with the present study.

Emission indices for $NO_x$ and $PM_1$ from inland water navigation have been used in emission inventories by Klimont et al. (2002), Rohacs and Simongati (2007), TNO (2008), CBS (2009) and UBA (2013). The authors reported $EI_{NO_x}$ and $EI_{PM_1}$ in the range 46-51 g $kg^{-1}$ and 1.5-4.0 g $kg^{-1}$, respectively (see table 1). From these data uncertainty($2\sigma$)-weighted average values for $EI_{NO_x}$ of $48\pm2$ g $kg^{-1}$ and $EI_{PM_1}$ $2.7\pm1.2$ g $kg^{-1}$ were derived, which are in a good agreement with the present study.

In order to comply with the limit values of the current RheinSchUO guideline for inland water navigation for $NO_x$ with 29-37 g kg$^{-1}$ a further significant reduction of the $NO_x$ emission is necessary. This can be achieved e.g. by using exhaust after-treatment systems, whose functional capability have been demonstrated in recent studies (Cooper, 2001, Schweighofer and Blaauw, 2009, BMVBS, 2012, Future Carrier, 2012, Hallquist et al., 2013, Pirjola et al., 2014). For example, the European project "The cleanest ship" (Schweighofer and Blaauw, 2009) shows that $NO_x$ and PM emission of a ship diesel engine equipped with an SCR (selective catalytic reduction) system and particle filter can be reduced to 4 g kg$^{-1}$ and 0.02 g kg$^{-1}$, respectively.

## 4 Summary and Conclusion

The present study has shown that the measurement site at the Rhine river provided representative real world emission data from inland navigation. Emissions of NO, $NO_2$, $CO_2$, and particulate matter from a large number of individual ships were monitored and analyzed. Particulate emissions measured in the ship plumes were dominated by PM1. An average $NO_2/NO_x$ emission ratio of 0.08±0.02 was obtained, which is typical for ship diesel engines without after-treatment systems such as oxidation catalysts or PM filter systems. The emission indices, emitted mass of pollutant per kg burnt fuel for $NO_x$ ($EI_{NO_x}$) and $PM_1$ ($EI_{PM_1}$) determined for different motor ship types (cargo, petroleum tanker and push tow) and for different operation parameters (L=loaded, U=unloaded, A=upstream and D=downstream) exhibited a large variation and were almost independent of the ship types and operation parameters. For the motor ship types and operation conditions investigated a weighted average $EI_{NO_x}$ of 54±4 g kg$^{-1}$ and lower limit $EI_{PM1}$ of ≥ 2.0±0.3 g kg$^{-1}$ was obtained with minimum and maximum values ranging from 20-161 g kg$^{-1}$ for $NO_X$ and ≥ 0.2-8.1 g kg$^{-1}$ for $PM_1$, respectively. The $EI_{NO_x}$ and $EI_{PM_1}$ from the present study are in a good agreement with literature data. The comparison of emission indices for $NO_x$ and $PM_1$ with limit values of the German Guidelines (BinSchAbgasV, 2005, RheinSchUO, 2011) showed that $NO_x$ emissions of all motor ship types investigated were above the limit values whereas the obtained lower limit $PM_1$ emissions for almost all motor ship types were just within the limit values. In order to meet the limit values for $NO_x$ und PM, in particular the $NO_X$ emissions have to be reduced significantly, e.g. by the introduction of specific exhaust after-treatment systems, some of which have been proven to be very effective.

Future campaigns should include PM size distribution and also CO, $SO_2$ and NMVOC measurements. Campaigns should be carried out at different seasons, to study a potential impact of water level and river streaming on the emissions.

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

**Figure Caption**

**Figure 1.**     Location of the measurement site at Rhine kilometre 843.
(This map is made available under the Open Database License: http://opendatacommons.org/licenses/odbl/1.0/. Any rights in individual contents of the database are licensed under the Database Contents License:
http://opendatacommons.org/licenses/dbcl/1.0/     -     See     more     at: http://opendatacommons.org/licenses/odbl/#sthash.hMw4LgYT.dpuf).

**Figure 2.**     Temporal variation of the NO, $NO_2$, $O_3$ and $CO_2$ concentration at the measurement site on February 20, 2013 from 11:30 to 14:00 from different ship types (G=goods ship, T=petroleum tanker, PT=push tow) and at different operation parameters (L=loaded, U=unloaded, A=upstream and D=downstream).

**Figure 3.**     Plot of $O_x$ vs. $NO_x$.

**Figure 4.**     Temporal variation of $CO_2$, $PM_{10}$ and $PM_1$ at the measurement site on February 20, 2013 from 11:50 to 12:10 for different ship types (G=goods ship, T=petroleum tanker) and different operation parameters (L=loaded, U=unloaded, A=upstream and D=downstream).

**Figure 5.**     Temporal variation of the NO, $NO_2$, $CO_2$ and $PM_1$ concentration and the integrated emission peaks as $\Delta NO$, $\Delta NO_2$, $\Delta CO_2$
and $\Delta PM_1$ peak area at the measurement site on February 20, 2013 from 11:50 to 12:10 for a goods ship (G) under loaded (L) and upstream (A) conditionsbed flow photo-reactor, with movable injector and turbulence barriers.

**Figure 6.**     $EI_{NO_x}$ (as $NO_2$) in g $kg^{-1}$ burnt fuel of single motor ships [goods] and the weighted average value of $EI_{NO_x}$ for different operation parameters, L=loaded, U=unloaded, A=upstream and D=downstream. Red bars show outliers ($4\sigma$ limit) and were not taken into account in the calculation of the weighted average value.

**Figure 7.**     Lower limit $EI_{PM_1}$ in g $kg^{-1}$ burnt fuel of single motor ships [goods] and the weighted average $EI_{PM_1}$ for different operation parameters, L=loaded, U=unloaded, A=upstream and D=downstream.

**Figure 8.** Weighted average emission index for $NO_x$ ($EI_{NO_x}$) in g kg$^{-1}$ burnt fuel for different motor ship types (G=goods, T=tanker and PT=push tow) at different operation parameters, (L=loaded, U=unloaded, A=upstream and D=downstream) in comparison with German guidelines (BinSchAbgasV, 2005 [yellow] and RheinSchUO, 2011 [green])

**Figure 9.** Weighted average lower limit emission index for $PM_1$ ($EI_{PM_1}$) in g kg$^{-1}$ burnt fuel for different motor ship types (G=goods, T=tanker and PT=push tow) at different operation parameters, (L=loaded, U=unloaded, A=upstream and D=downstream) in comparison with German guidelines (BinSchAbgasV, 2005 [yellow] and RheinSchUO, 2011 [green]).

**Figures**

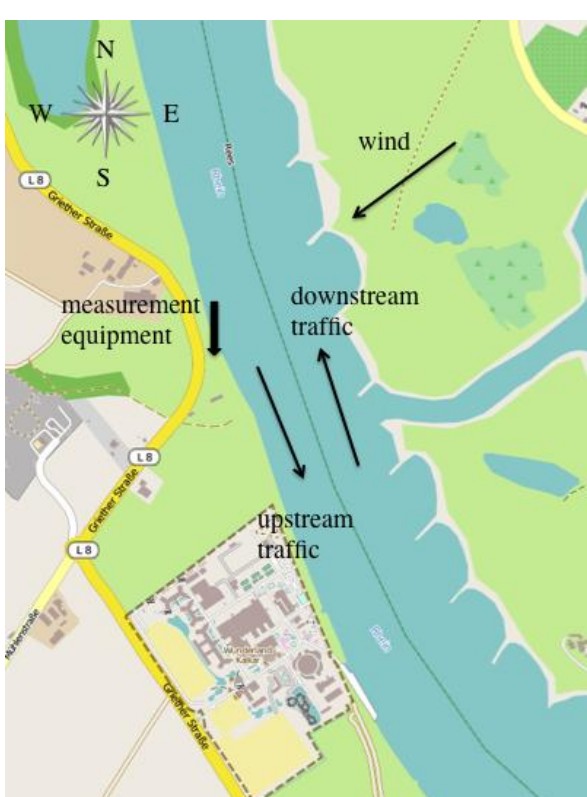

**Figure 1.**

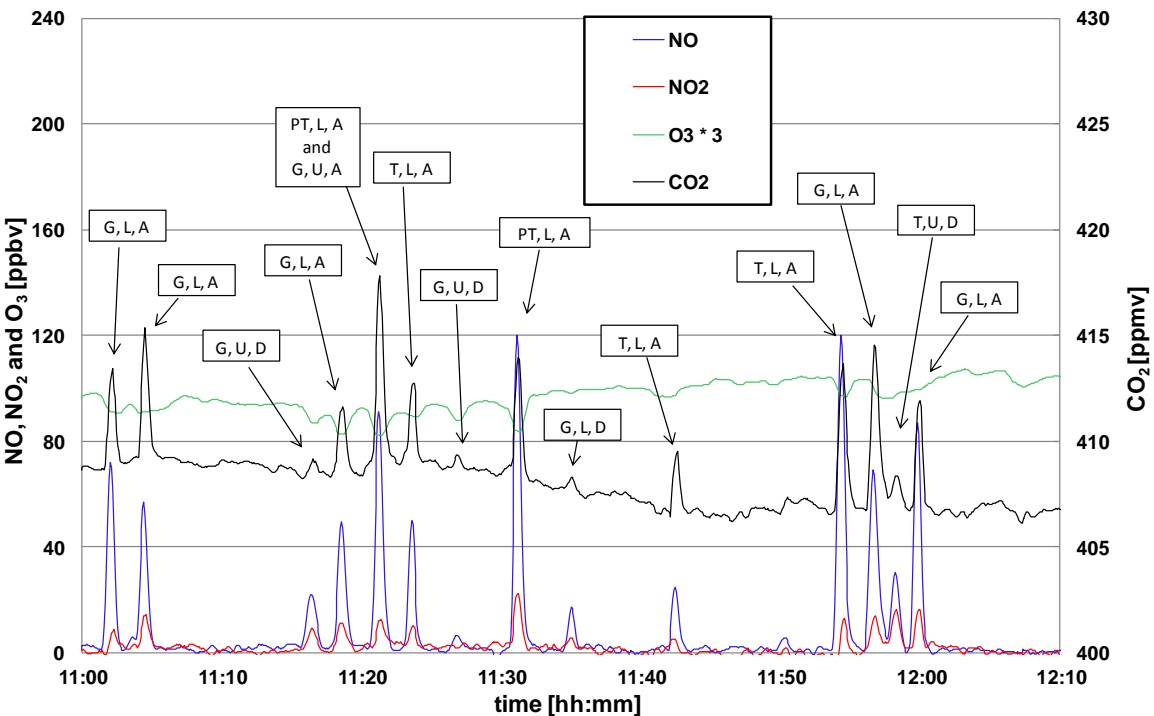

**Figure 2.**

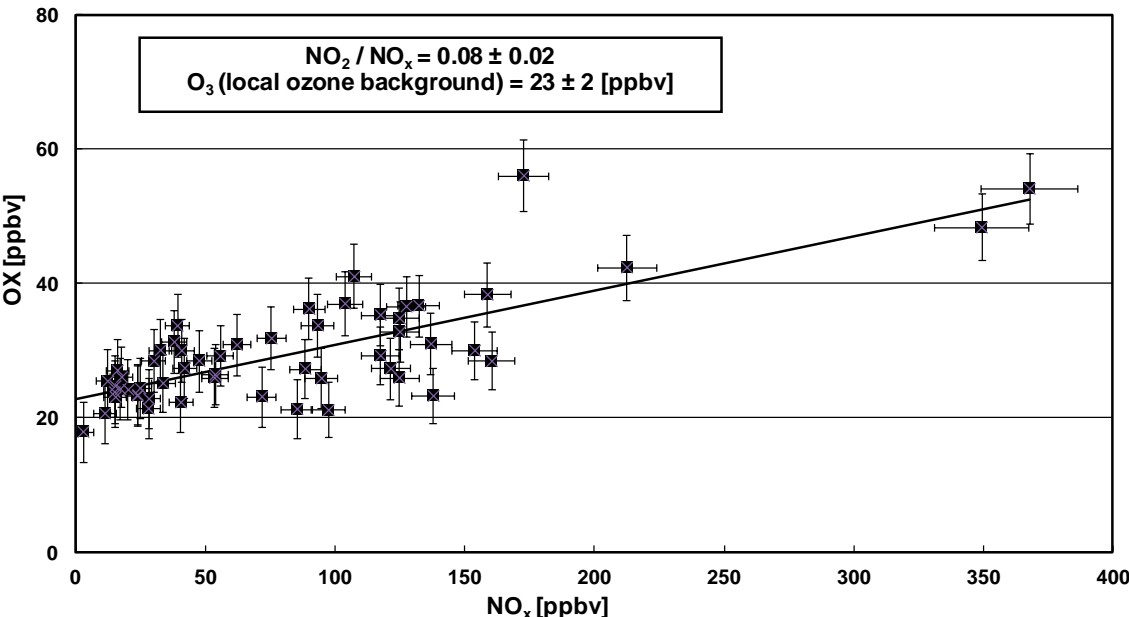

**Figure 3.**

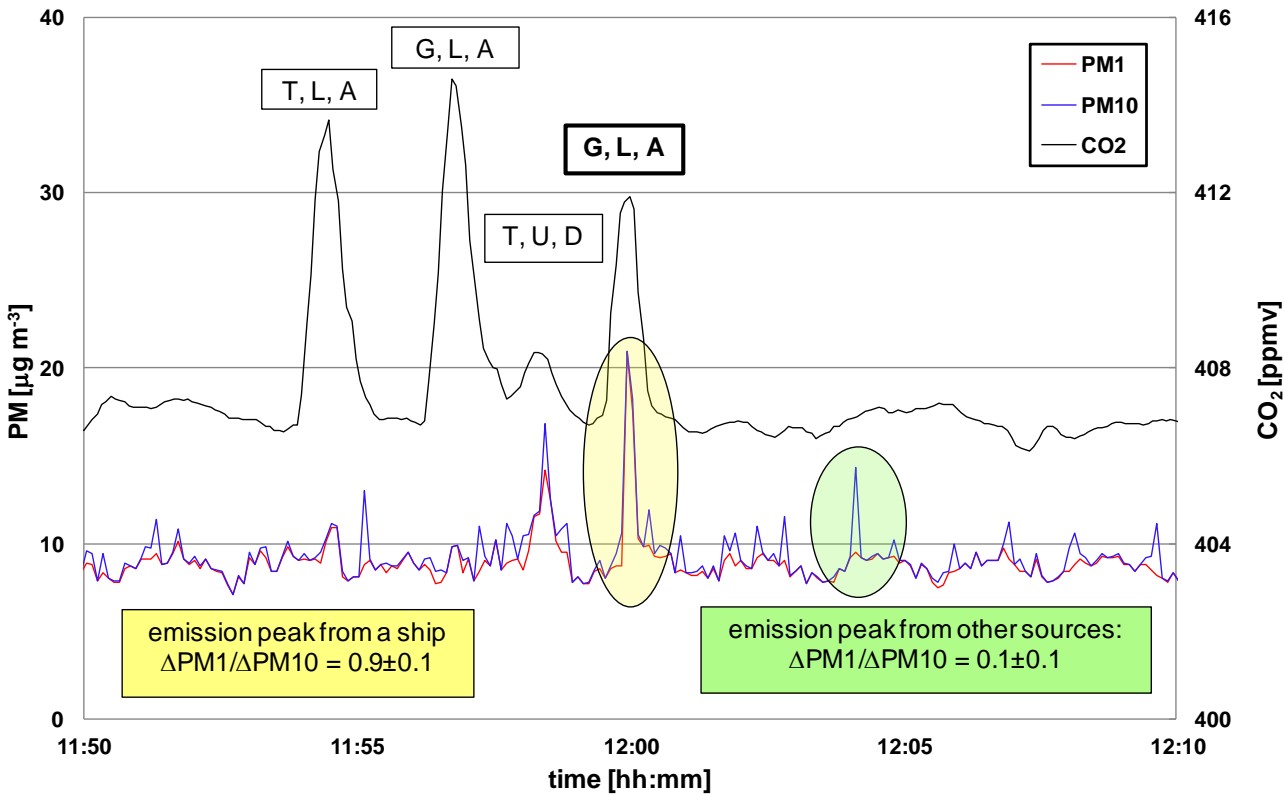

**Figure 4.**

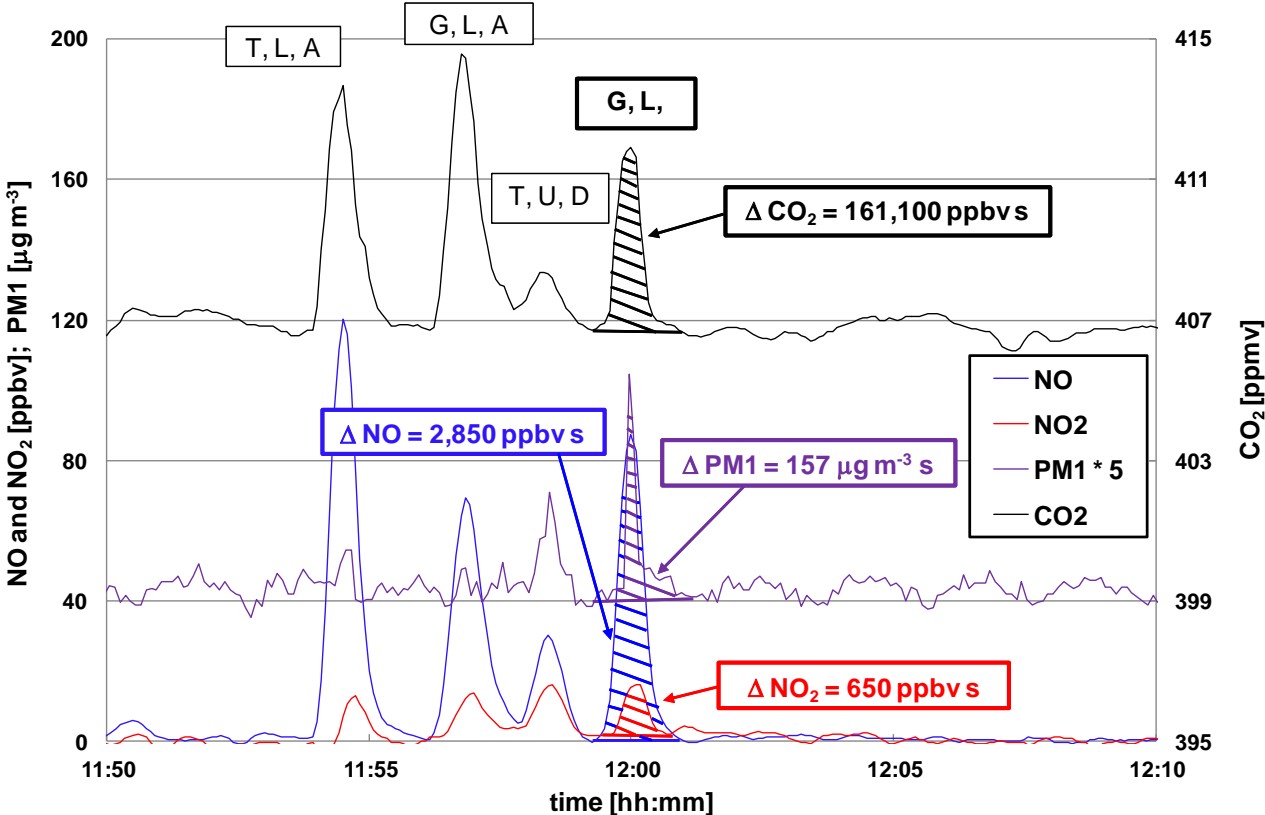

**Figure 5.**

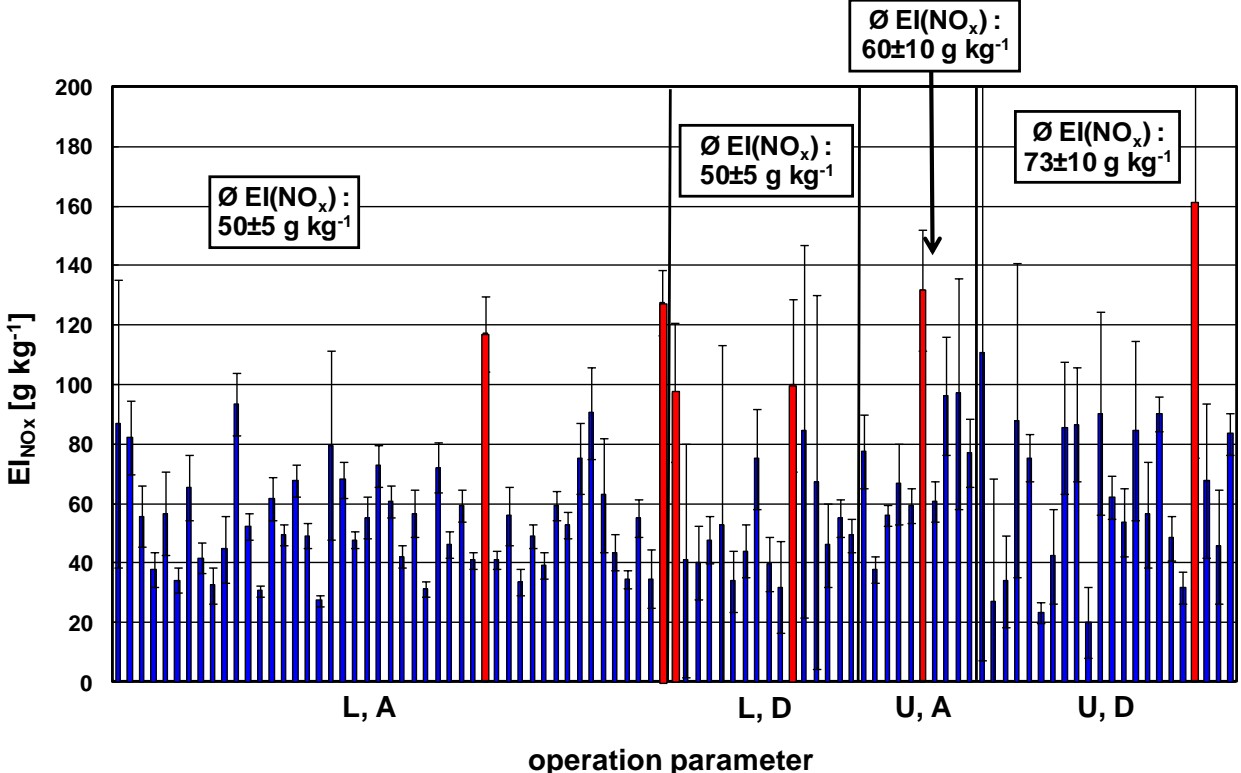

**Figure 6.**

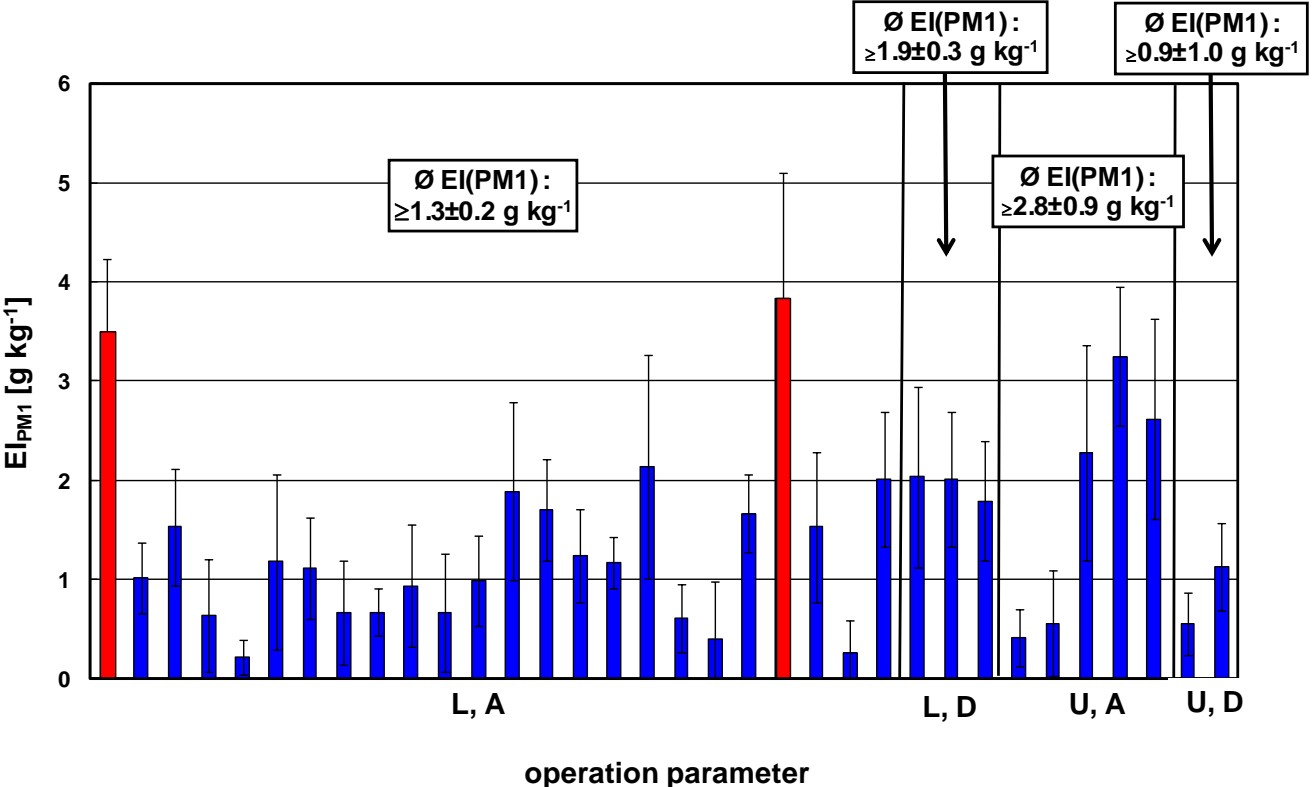

**Figure 7.**

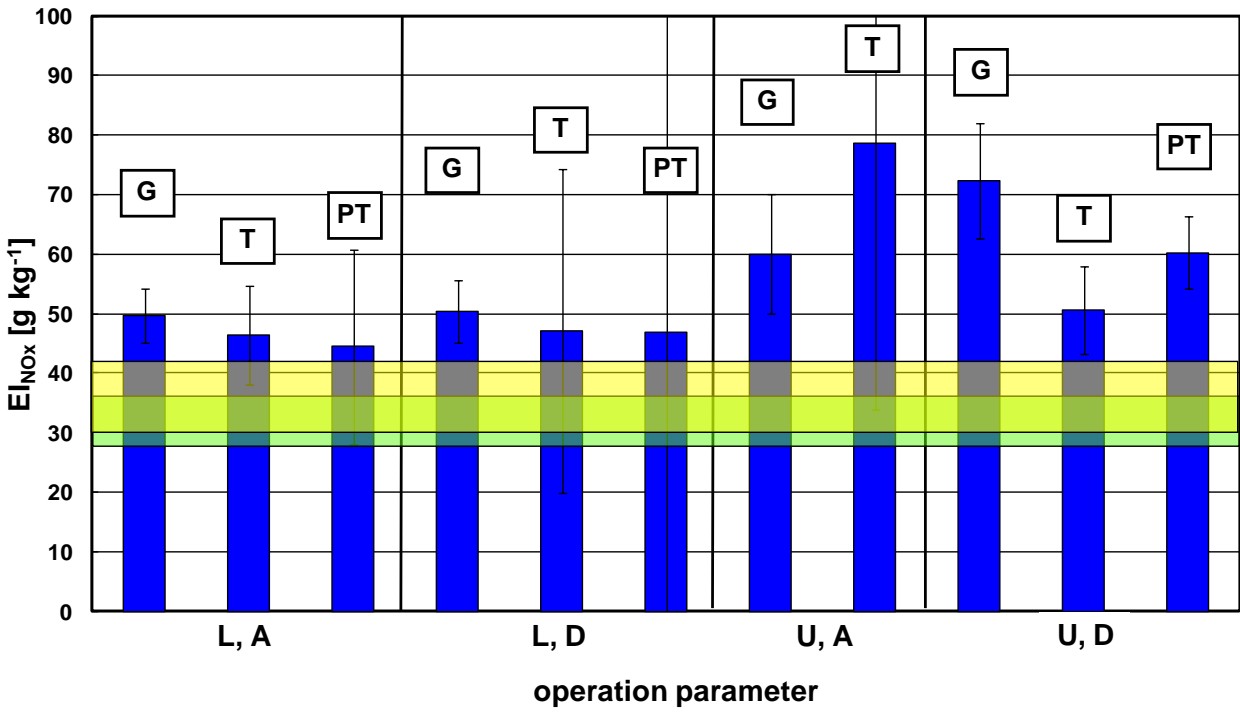

**Figure 8.**

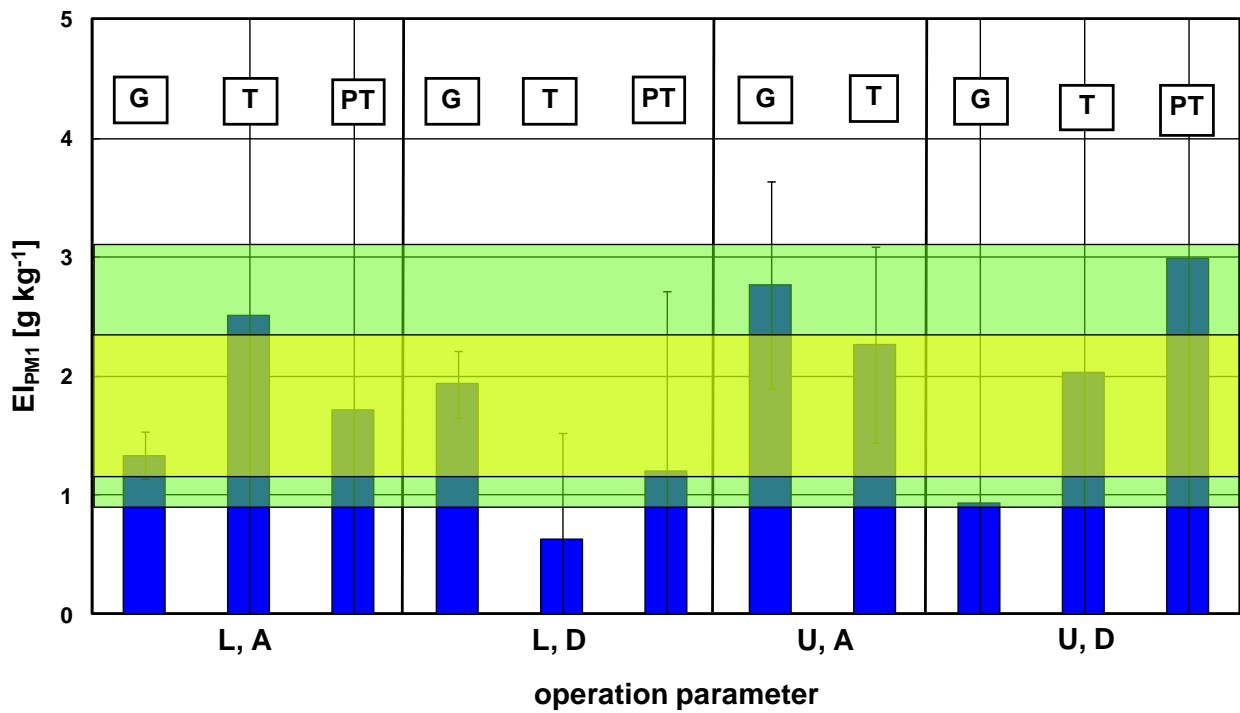

**Figure 9.**

**Table Caption**

**Table 1.**     Emission indices $NO_x$ and $PM_1$ in g kg$^{-1}$ burnt fuel calculated from the measured values in comparison with different literature data from inland water transportation.

| Reference | Location | Sampling period | $EI_{NO_x}$ [g kg$^{-1}$] | $EI_{PM_1}$ [g kg$^{-1}$] | Ship types |
|---|---|---|---|---|---|
| **A) field measurements (inland, engine without exhaust gas after-treatment system)** | | | | | |
| This study | Germany, Rhine (inland) | 2013 | 54 ± 4 | ≥ 2.0 ± 0.3 | different |
| Kesgin and Vardar (2001) | Turkey; Bosporus (inland) | 1998 | 57 | 1.2 | domestric passenger ships (a) |
| Trozzi and Vaccaro (1998) | Italy, Tyrrhenian Sea (inland) | 1998 | 51 | 1.2 | domestric passenger ships (a) |
| Van der Gon and Hulskotte (2010) | Netherlands (inland) | 2010 | 45 | 1.9 | different |
| Schweighofer and Blaauw (2009) | inland | 2009 | 39 | 0.73 | research vessel (b) |
| **B) field measurements (inland, engine with exhaust gas after-treatment system)** | | | | | |
| BMVBS (2012) | inland | 2011 | n.d. | 0.08 – 0.48 | research vessel |
| Futura Carrier (2010) | inland | 2009 | n.d. | 0.29 ± 0.01 | research vessel |
| Schweighofer and Blaauw (2009) | inland | 2009 | 11 - 39 | 0.02 | research vessel (c) |
| **C) inventories** | | | | | |
| Rohacs and Simongati (2007) | Average EU (inland) | 2007 | 47 | 3.2 | inventory |
| TNO (2008), CBS (2009) | Netherlands (inland) | 2008-2009 | 46 | 1.9 | inventory |
| Klimont et al. (2002) | RAINS, EU (inland) | 2002 | 51 | 4.0 | inventory |
| UBA (2013) | TREMOD, Germany (Inland) | 2013 | 49 ± 6 | 1.5 ± 0.2 | inventory |

Remarks: n. d. no data, a) domestric passenger ships with diesel engine (medium-speed), b) without exhaust gas after-treatment system, c) with exhaust gas after-treatment system

**Table 1.**

