# Peer review of "Emissions of NO, NO2, and PM from inland shipping"

_Atmospheric Chemistry and Physics, 2016_

## Referee Comment (RC1) · Anonymous Referee #1 · 2 Aug 2016

Inland water transport emissions of NOx, CO2 and PM were investigated under real world conditions at the river Rhine, Germany during a campaign from February 20, to February 22, 2013. An average NO2/NOx emission ratio of  $0.08 \pm 0.02$  was determined, which is indicative of ship diesel engines without after-treatment systems. The NOx emissions of all investigated motor ship types (cargo, petroleum tanker and push tow) and for different operation parameters (L=loaded, U=unloaded, A=upstream and D=downstream) were above the threshold values of national German guidelines, while the obtained PM1 emissions were just within. The implementation of after-treatment systems is recommended for reduction of NOx emissions to acceptable values in relation to national German guidelines. 1 Introduction General comments According to the European Commission's White Paper from 2011 an increasing part of road freight transported over more than 300 km distance should shifted to other transport modes

such as waterborne or rail transport. This will reduce the direct impact on climate change. Following this development the knowledge of the corresponding NOx, VOC, PM and CO emission variations that means environmental impact is important. This paper contributes more real-world information about this topic. The paper addresses relevant scientific questions within the scope of ACP. The paper presents novel concepts, ideas and tools. The scientific methods and assumptions are valid and clearly outlined so that substantial conclusions are reached. The results are sufficient to support the interpretations and conclusions. The description of experiments and calculations are sufficiently complete and precise to allow their reproduction by fellow scientists. The authors should more clearly indicate their own new/original contribution and the representativeness of their results (Why a three-days campaign is representative?). The quality and information as well as the captions of the figures are good. But the captions of some figures and tables should be improved so that these are understandable without the text. The related work is well cited. The number and quality of references is appropriate. Title and abstract reflect the whole content of the paper. The overall presentation is well structured and clear. The language is fluent and precise. The mathematical symbols, abbreviations, and units are generally correctly defined and used. Specific Comments NO, NO2, O3, CO2, PM1 and PM10 concentrations were measured together with wind speeds and wind directions so that NO2/NOx emission ratios and emission indices of NOx and PM1 can be calculated. The correct NO2/NOx ratio is obtained by plotting Ox, which is the sum of NO2 and O3 versus the measured NOx concentration. Errors were calculated using error propagation for the different measured compounds. An outlook is missing to address open questions as e.g. the influences of water level and river streaming velocity. Why NMVOCs were not detected to find carcinogenic substances? Technical corrections For some reports the institution, city and country are missing in the references. In some references a "." should be set instead of a "," after the paper title.

---

## Referee Comment (RC2) · P. Sturm (Referee) · 15 Aug 2016

The paper is of interest as it tackles real world emissions of an emission source which is not easy to be accessed. There are some minor changes requested. In general I think a change of the title from "... inland water transportation" to "..inland shipping" would be appropriate A further general remark is the usage of after-treatment systems. It should be replaced by "exhaust gas after-treatment systems". In the abstract the EIs are given in g per kg. The term "fuel burnt" is missing, i.e. g per kg fuel burnt. Measurement setup: there is some text about the instruments used, however a clear description about the set-up is missing. Measurement height for pollutants, wind etc.

Remark to Figure 4: avoid the term "immission" Figure 8 and 9 contain the classification "G", "T", "TP". Although being explained in the caption, there is no reference to them

in the text.

---

## Author Response (AR1)

**Reply to Referee 1:**

We are quite grateful for the referee's comments, which are generally very positive. Because of the limited funding, it was unfortunately not possible to study also NMVOC emissions from ships. Hopefully the publication of our results will help to convince funding agencies either in Germany or the EU that our approach generates very useful data on shipping emissions and that we will be able to monitor also NMVOC emissions in the future with additional funding. This is also the reason why we performed only a three-day campaign. However, we were able to study a much larger number of ships compared with other investigations, so that we are convinced that our study is representative. A sentence about possible future campaigns has been added to the section "Summary and Conclusion".

References have been thoroughly checked and errors have been corrected.

**Reply to Referee Peter Sturm:**

We are quite grateful for Peter Sturm's comments, which helped us improving our manuscript. Below are our answers to the referee's comments:

Reviewer:

1. In general I think a change of the title from"…inland water transportation" to ..inland shipping"
- We agree. The title of the manuscript has been modified accordingly.

2. The usage of after-treatment systems should be replaced by "exhaust gas after-treatment systems"
- The manuscript has been changed accordingly.

3. In the abstract the EIs are given in g per kg. The term "burnt fuel" is missing.
- The manuscript has been changed.

4. Measurement setup. There is some text about the instruments used, however a clear description about the set-up is missing. Measurement height for pollutants, wind etc."
- A sentence has been added in which the measurement height for the pollutants is given.

5. Remarks to Figure 4: avoid the term "immission"
- Manuscript has been changed accordingly.

5. Figure 8 and 9 contain the classification "G", "T", "TP". Although being explained in the caption, there is no reference to them in the text.
- Explanation has been added to the manuscript section "Results and Discussion".

**Marked-up manuscript:**

[revised manuscript text omitted]

**Table Caption**

**Table 1.** Emission indices $NO_x$ and $PM_1$ in g $kg^{-1}$ burnt fuel calculated from the measured values in comparison with different literature data from inland water transportation.

| Reference | Location | Sampling period | $\mathbf{EI_{NO_x}}$ [g kg$^{-1}$] | $\mathbf{EI_{PM_1}}$ [g kg$^{-1}$] | Ship types |
|---|---|---|---|---|---|
| **A) field measurements (inland, engine without exhaust gas after-treatment system)** | | | | | |
| This study | Germany, Rhine (inland) | 2013 | 54 ± 4 | ≥ 2.0 ± 0.3 | different |
| Kesgin and Vardar (2001) | Turkey; Bosporus (inland) | 1998 | 57 | 1.2 | domestric passenger ships (a) |
| Trozzi and Vaccaro (1998) | Italy, Tyrrhenian Sea (inland) | 1998 | 51 | 1.2 | domestric passenger ships (a) |
| Van der Gon and Hulskotte (2010) | Netherlands (inland) | 2010 | 45 | 1.9 | different |
| Schweighofer and Blaauw (2009) | inland | 2009 | 39 | 0.73 | research vessel (b) |
| **B) field measurements (inland, engine with exhaust gas after-treatment system)** | | | | | |
| BMVBS (2012) | inland | 2011 | n.d. | 0.08 – 0.48 | research vessel |
| Futura Carrier (2010) | inland | 2009 | n.d. | 0.29 ± 0.01 | research vessel |
| Schweighofer and Blaauw (2009) | inland | 2009 | 11 - 39 | 0.02 | research vessel (c) |
| **C) inventories** | | | | | |
| Rohacs and Simongati (2007) | Average EU (inland) | 2007 | 47 | 3.2 | inventory |
| TNO (2008), CBS (2009) | Netherlands (inland) | 2008-2009 | 46 | 1.9 | inventory |
| Klimont et al. (2002) | RAINS, EU (inland) | 2002 | 51 | 4.0 | inventory |
| UBA (2013) | TREMOD, Germany (Inland) | 2013 | 49 ± 6 | 1.5 ± 0.2 | inventory |

Remarks: n. d. no data, a) domestic passenger ships with diesel engine (medium-speed), b) without exhaust gas after-treatment system, c) with exhaust gas after-treatment system

**Table 1.**